# Magnetic Immunosensor Coupled to Enzymatic Signal for Determination of Genomic DNA Methylation

**DOI:** 10.3390/bios12030162

**Published:** 2022-03-04

**Authors:** Yitao Liang, Bin Zhang, Zexin Xue, Xuesong Ye, Bo Liang

**Affiliations:** 1College of Biomedical Engineering and Instrument Science, Zhejiang University, Hangzhou 310027, China; liang1tao@zju.edu.cn (Y.L.); 3170102110@zju.edu.cn (Z.X.); yexuesong@zju.edu.cn (X.Y.); 2Sir Run-Run Shaw Hospital, School of Medicine, Zhejiang University, Hangzhou 310016, China; doctorbinzhang@zju.edu.cn

**Keywords:** DNA methylation, magnetic beads, screen printed electrode, electrochemical

## Abstract

Aberrations of genomic DNA methylation have been confirmed to be involved in the evolution of human cancer and have thus gained the potential to be depicted as biomarkers for cancer diagnostics and prognostic predictions, which implicates an urgent need for detection of total genomic DNA methylation. In this work, we suggested an assay for the quantification of global DNA methylation, utilizing methylation specific antibody (5mC) modified magnetic beads (MBs) for immunorecognition and affinity enrichment. Subsequently, the captured DNA on the surface of MBs interacted with the glucose oxidase-conjugated DNA antibody whose catalytic reaction product was engaged in electrochemical detection of the overall level of DNA methylation on a PB-doped screen-printed electrode. With 15 pg of input DNA, which, to our best knowledge, is the lowest required amount of DNA without sodium bisulfite treatment or amplification, this test strategy was able to perceive as low as 5% methylation level within 70 min including the preparation of anti-5mC-MBs. We believe this detection technique offers a promising option to detect global DNA methylation in both academic and clinical scenarios.

## 1. Introduction

As one of the major epigenetic modifications of DNA, DNA methylation, which is that a methyl group (-CH_3_) was added to the fifth carbon of a cytosine following a guanine nucleotide (CpG sites), is known to be crucial in gene expression regulation and thus involved in various cellular processes, including development and disease [1,2,3,4]. Alteration of 5-methylcytosine (5mC) at global levels is more resistant than that in selected loci [5]. Therefore, genome-wide DNA methylation determination is gaining prominence as prognostic and diagnostic biomarkers for clinical application depicting early neoplasia and solid tumors and being strongly associated with tumor aggressiveness [6,7,8,9], regardless of cancer tissue origin [10,11,12].

There are mainly three categories of strategies utilized to distinguish methylated from unmethylated DNA—sodium bisulfite conversion, methylation-sensitive restriction enzymes (MSREs) and affinity-based techniques. PCR amplification step is necessary after bisulfite conversion and incomplete conversion lead to false-positive results [13,14]. Sites that can be analyzed are highly confined by the cleavage site of available MSREs and they may not be able to identify other epigenetic modifications of DNA, such as 5-hydroxymethylcytosine (5hmC) [15]. On the other hand, generally, it is not required to have the knowledge of specific sequences in order to determine the global DNA methylation level. Enzyme-linked immunosorbent assay (ELISA) is one of the most commonly used technique to assess global DNA methylation. Antibodies against 5mC show better purification performance of denatured single-stranded DNA (ssDNA) with sparse CpGs whilst methyl-CpG-binding domains (MBDs) tend to interact double-stranded DNA (dsDNA) and are biased to the enrichment of CpG islands (CGIs) [16,17,18].

Recently, electrochemical biosensors have exhibited several dramatic features in terms of high sensitivity and specificity, cost-effectiveness, field portability, simple operation and handling, compatibility with miniaturization and short response time [12]. To deal with the low amounts of methylated DNA in a clinical sample [19], a few strategies have been exploited to concentrate the analyst. Immuno-magnetic beads (MBs) afford a quite attractive characteristic in aspects of sensitivity because of their ability to achieve high loadings of biomolecules in solution under constant stirring [20,21,22]. Enrichment of methylated DNA with anti-5mC coated MBs has been successfully demonstrated by another research group [23,24,25]. As for detection signal, multiple enzymes are applied as tracer labels in the development of electrochemistry based assay, e.g., horseradish peroxidase (HRP) [26], glucose oxidase (GOx) [27,28]. Incorporation of electron-transfer-mediators (ETM) at the surface of the electrode improves electron communication between the redox center of biomolecules and the electrode. Among all the available options, Prussian blue (PB) has been utilized extensively due to its excellent properties as a transducer platform for hydrogen peroxide (H_2_O_2_) quantification [29,30,31,32]. The electrocatalytic ability of PB towards H_2_O_2_ reduction can be induced at a very low voltage, which offers specificity to H_2_O_2_ detection.

Herein, this paper reports a method for determination of 5mC at a global level based on screen printed PB-doped carbon electrode with engagement of electrochemical enzymatic signal (Figure 1) without either bisulfite treatment or PCR amplification. In this methodology, streptavidin MBs were firstly incubated with a biotinylated anti-5mC antibody. Then, the constructed anti-5mC-MBs were mixed with the extracted genomic DNA which was denatured at 95 °C in advance so as to make it more efficient for the antibody to capture the target methylated DNA. After the immune-recognition of captured DNA on the surface of MBs with GOx conjugated DNA antibody (anti-DNA-GOx), the MBs was incubated with glucose where the GOx worked as a catalyst. Catalysate within the supernatant was pipetted onto the screen printed PB-doped carbon electrode after magnetic separation. Finally, chronoamperometric (CA) response is directly proportional to the amount of H_2_O_2_ (i.e., the amount of methylation level in the extracted DNA) on the surface of electrode, and this analysis technique has been experimented with in testing the methylation level in three different hepatocellular carcinoma (HCC) cell lines.

## 2. Materials and Methods

### 2.1. Reagents and Chemicals

Streptavidin Magnetic Beads (1 μm diameter, 10 mg mL^−1^, Cat. No: HY-K0208) were purchased from MedChemExpress, Monmouth Junction, NJ, USA; biotinylated mouse anti-5-methylcytosine monoclonal antibody (Cat. No: ab179914) and glucose oxidase conjugation kit (Cat. No: ab102887) were purchased from Abcam, Waltham, MA, USA; mouse anti-DNA monoclonal antibody (Cat. No: 7115) was purchased from Chondrex, Inc., Woodinville, WA, USA; goat anti-mouse IgG secondary antibody, cyanine3 were purchased from Invitrogen (Cat. No: A10521), Waltham, MA, USA; Tween-20 was purchased from Sigma-Aldrich, St. Louis, MO, USA; Tris-EDTA (TE) buffer was purchased from Macklin, Shanghai, China; phosphate buffer saline (PBS) buffer, 20X SSC buffer, bovine serum albumin (BSA) were purchased from Sangon Biotech, Shanghai, China; ELISA washing buffer was purchased from Beyotime, Shanghai, China; Tris-HCl buffer was purchased from Aladdin, Shanghai, China; glucose was purchased from Sinopharm Chemical Reagent Co., Ltd., Beijing, China. PBS contains 10 mM Na_2_HPO_4_, 0.137 M NaCl, 2.7 mM KCl, 1.8 mM KH_2_PO_4_; 5X SSC buffer comprised 0.75 M NaCl, 75 mM sodium citrate; Tris buffer comprised 50 mM tris(hydroxymethyl)aminomethane. GOx-conjugated DNA antibody was prepared according to the manufacturer’s procedure.

### 2.2. Preparation of PB-Doped Carbon Electrode

The fabrication process of the screen-printed electrodes on the polyethylene terephthalate (PET) substrate was similar to our previous work [32,33]. Briefly, Ag/AgCl paste, PB-doped carbon ink (Gwent Group Ltd., Newport, UK) and isolated oil (Jujo Chemical Co., Ltd., Tokyo, Japan) were successively printed onto the substrate after ultrasonic cleaning process with ethanol and deionized water. The substrate was heated for 30 min at 60 °C to let the paste dry and eliminate residual solvent after each printing step. Finally, individual SPE was formed by cutting the screen-printed film. Altogether, the SPE consisted of two electrodes: a PB-doped square working electrode (side length: 2 mm), and an Ag/AgCl electrode as reference electrode.

### 2.3. Extraction of Genomic DNA

Three different hepatocellular carcinoma (HCC) cell lines (SNU-449, SK-Hep-1, Hep G2) were kindly provided by the Key Laboratory of the Laparoscopic Technology of Zhejiang Province, Department of General Surgery, Sir Run-Run Shaw Hospital. CpG Methylated Jurkat Genomic (100% methylated) DNA was purchased from Thermo Fisher, Waltham, MA, USA. Preparation of whole genomic amplified (WGA) was done by following the protocol of REPLI-g whole genome amplification mini kit (Qiagen, Hilden, Germany), and WGA was considered to be 0% methylated as negative control.

### 2.4. MBs Modification

One point five μL of 10 mg/mL Streptavidin MBs (MedChemExpress) suspensions were deposited in a 1.5 mL Eppendorf tube. Firstly, MBs were washed twice with Buffer consisting of PBS, pH 7.4 and 0.05% Tween-20 before being incubated with 15 μL biotinylated 5mC antibodies for 30 min (37 ℃, 75 rpm, rotate and shake). After performing two washing steps with Buffer, the anti-5mC-MBs were resuspended with Buffer and stored at 4 °C for later use. Target DNA solution was denatured at 95 °C for 10 min and transferred immediately to ice for 3 min followed by dilution with 5X SSC buffer containing 1% BSA to obtain the desired concentration of single-stranded DNA. BSA was used to block uncovered MBs’ surface area with target DNA so as to minimize the non-specific adsorption of biological molecules. Subsequently, anti-5mC-MBs were incubated with target DNA solution for 15 min (37 °C, 75 rpm, rotate and shake). After two washing steps were carried out with 1X ELISA washing buffer, the DNA-MBs were incubated with anti-DNA-GOx solution (dil. 1/100 in 5X SSC and 1%BSA buffer) for 15 min (37 °C, 75 rpm, rotate and shake). A 3D rotating mixer (Yooning Instruments, Hangzhou, China) was used for homogenization of the MBs suspensions.

### 2.5. Chronoamperometric Detection

Fifteen μL of GOx-MBs went through magnetic separation in an EP tube before 15 μL background solution (BGS) containing 10 mM glucose in Tris-HCl (50 mM, pH = 7.0) were added into the tube. The suspensions were incubated at 37 °C for 5 min before the supernatant pipetted onto the SPE to perform amperometric measurements. Since H_2_O_2_ is electroactive, the amount of enzymatically generated H_2_O_2_ was measured chronoamperometrically by applying the potential of −0.2 V vs. Ag/AgCl pseudoreference electrode with a potentiostat (CompactStat.h, IVIUM technologies, Eindhoven, The Netherlands) for 100 s.

## 3. Results and Discussion

### 3.1. Detection of H_2_O_2_ Using PB Electrodes

Before screen printed PB electrodes were applied to detect the product generated from oxidase catalyzed reactions, their catalytic activity towards H_2_O_2_ was explored. The concentration range of H_2_O_2_ reduction test conducted at the surface of PB electrode was chosen as 0–0.5 mM because the quantity of H_2_O_2_ produced by GOx was deemed to be small. Figure 2a demonstrates that the amplitude of reduction current increased with the H_2_O_2_ concentration from 0 mM to 0.5 mM, which verifies the electrocatalytic reduction of H_2_O_2_. Figure 2b illustrated that amperometric responses (Figure 2a and Appendix A) of PB electrode were taken at 100 s by applying four different test potentials and the acquired current change in response to increasing H_2_O_2_ concentration. Figure 2c displays the sensitivity of the PB electrode obtained within the concentration range at different potentials. PB-doped electrodes (*n* = 3) exhibited a relatively low sensitivity of −144 nA/mM with SD of 4 nA/mM at 0 V, and at −0.1 V, the sensitivity was higher, which was −802 nA/mM with SD of 3 nA/mM. The sensitivity at −0.2 V, which was −1.15 × 10^3^ nA/mM with SD of 2 × 10^1^ nA/mM, was similar but with less standard deviation when compared to that at −0.25 V, −1.14 × 10^3^ nA/mM with SD of 3 × 10^1^ nA/mM, indicating that this homemade PB electrode had good electrocatalytic ability towards H_2_O_2_ at −0.2 V, which was selected as the applied potential for later chronoamperometric measurements.

### 3.2. Selectivity of the Assay

Figure A2 illustrated the successful immobilization of anti-5mC onto the MBs with immunofluorescence. What is more, the physical structures of the MBs surface were shown in Figure A3. Affinity interaction between the specific antibody and the target analyte enhanced the selectivity of this detection method. Investigations were performed in order to verify the assay specificity with equal quantity (15 pg) of CpG Methylated Jurkat Genomic (100% methylated) and unmethylated (WGA) DNA while the biorecgonition events happened at the surface of anti-5mC-MBs. After labeling MBs with anti-DNA-GOx and incubating them in glucose solution, the CA responses of the suspensions for both samples were measured, respectively. Figure 3a reveals the different reduction current of two samples at 100 s (Jurkat is −220 nA and WGA is −38 nA in Figure 3b) and the CA responses for these samples were calculated by subtracting the background signal of glucose solution with the same electrode before sample tests. The difference in CA response between these two samples was considered significant (larger than 5.7 times). Results from two additional sets of experiment in which Tris-HCl took the place of target DNA (tagged as NoT) as well as a positive control group without anti-DNA-GOx (tagged as control) make further verification. CA responses of both NoT (−25 nA) and control (−33 nA) were closed to that of WGA. Given the control studies, it was strongly implicated that the presence of both methylation in the sample and anti-DNA-GOx were highly required to the assay response and when taken together with the response from synthetic samples, the feasibility of this assay to detect DNA methylation was further validated.

### 3.3. Assay Optimization

It is essential to optimize some experimental conditions for the purpose of getting the optimal analytical capability of this assay. As the volume of the solution dropped on the SPE for detection was limited to 15 μL based on its structure, the concentration of nucleic acid between 10^−6^ to 10^−1^ ng/μL was experimented with (Figure A4). The resulted CA responses increased with higher applied methylated nucleic acid concentration albeit with larger variance, so the total nucleic acid concentration was chosen as the medium within the range, 10^−3^ ng/μL.

Both incubation time and concentration of antibody jointly impact the affinity interaction; with the fixed concentration of antibody (dil. 1/100), incubation time was the one left to be optimized. There were three successive incubation steps in the TE tube: nucleic acid enrichment with anti-5mC-MBs (Figure 4a), then labeling the derived bioconjugates with anti-DNA-GOx (Figure 4b) and finally, the enzymatic reaction with glucose (Figure 4c). The influence of incubation time in all three procedures were optimized by evaluating amperometric responses obtained in the absence (WGA, 10^−3^ ng/μL) and presence of target methylated Jurkat DNA (10^−3^ ng/μL). For the construction of sandwich immuno recognition on the surface of MBs (Figure 4a,b), the resulted current increased with longer incubation time up to 15 min in the presence of target DNA, while in the absence of target DNA, the amperometric responses remained stable and nonsignificant. The GOx-labeled MBs and glucose were incubated for a certain period, which made it possible for the accumulation of enzymatical production, H_2_O_2_, within the suspension. Similar strategies have been employed in the microfluidic biosensors as so-called stop-flow protocol [28,34]. The choice of incubation period for glucose was 5 min as the CA responses did not change much ever after.

### 3.4. Amperometric Detection of Global DNA Methylation

By mixing the same concentration (10^−3^ ng/μL) of methylated Jurkat (100% methylated) and WGA (0% methylated) DNA with different volume ratio, preparation of 0–100% methylated DNA was achieved. Methylation level was determined by CA responses (Figure A5) which signified a linear relationship between reduction current and the total genomic content of DNA methylation. The derived calibration plot in Figure 5a illustrated that the linear regression equation was estimated to be *I* (current, nA) = −194(% methylation)–31 with a correlation coefficient R^2^ of 0.965 (*n* = 3) and standard errors of the slope and intercept in the equation were 13 and 8, respectively, which indicates that the CA response correlated to methylation level from 5% to 100%. In respect of reproducibility, relative standard deviation (%RSD) were calculated from the results of three independent detections, and they were to be <5.0% in this technique.

The analytical characteristics of this assay for global DNA methylation were quite competitive when compared with previously reported work with electrochemical technique (Table 1). Though direct affinity interaction between DNA samples and bare gold interface deformalizes the fabrication of biosensors [17,35,36], these methods required at least 25 ng of total input DNA while ours only cost 15 pg which much surpassed theirs. Besides, anti-5mC-MBs conduce to both specific recognition and enrichment of the target at the same time, and the second antibody conjugated with GOx intended to selectively label the MBs that captured target DNA. Additionally, the enzymatic reaction product accumulated within the suspension by incubation for a period which was also dedicated to lower the limit of detection (LOD). With regard to assay time, it took around 70 min in this work, which did not fall very far behind the gold-DNA affinity tactics (1–2 h).

### 3.5. DNA Methylation Analysis in Cell Lines

Three different HCC cell lines (SK-Hep-1, Hep G2 and SNU-449) were used to evaluate the applicability of this method. After DNA extraction and purification from these cell lines, samples were stored at –20 °C before detection. With an equal quantity (15 pg) of methylated Jurkat (100% methylation) and unmethylated WGA (0% methylation) as internal controls, the CA response of all three cell lines fell in the range of these two groups (Appendix A). Variations of CA responses from these cancer cell lines exhibit the presence of a quite different methylation level (current response for SK-Hep-1, Hep G2 and SNU-449 was about –201, –106 and –66 nA respectively). With the linear regression equation obtained in Section 3.4, the level of CA response suggests that SK-Hep-1 possesses more than 80% of methylation, while Hep G2 exhibits less than 40% methylation level and SNU-449 has a much lower methylation level, which is 20%, than that of the other two cancer cell lines (by comparing data in Figure 5a,b).

## 4. Conclusions

To summarize, with the help of anti-5mC coated MBs which could specifically recognize methylated DNA from the background with unmethylated DNA samples via affinity interaction and after labeling MBs with a second antibody conjugated with GOx, a screen printed PB-doped carbon electrode was used to detect the enzymatically generated H_2_O_2_ via chronoamperometry so as to quantify the global DNA methylation level as low as 5% methylation differences with 15 pg DNA input in 70 min.

There is still room for improvement of the detection sensitivity of the PB-doped SPE towards H_2_O_2_ and then the LOD of the assay can be even lower. Since previous findings have corroborated that the PB electrode possessed the catalytic activity towards electrochemical reduction of H_2_O_2_ which is generated by the oxidase-catalyzed reaction in this work. Hence, with the use of carbon mediator paste containing PB for SPE, the suggested assay did not require any further electrode modification steps after the completion of screen printing but possesses stability compared with the PB electrode modified by electrodeposition, and the chronoamperometry measurement was carried out at low working potential to ensure high sensitivity. Moreover, the volumes of solution in each step required to manipulate the MBs and for detection on the SPE are all quite small (about 15 μL); let alone that it is easy to attain and store the detection solution comprising merely glucose in the scenario of POCT exhibiting features such as low cost and portability.

## Figures and Tables

**Figure 1 biosensors-12-00162-f001:**
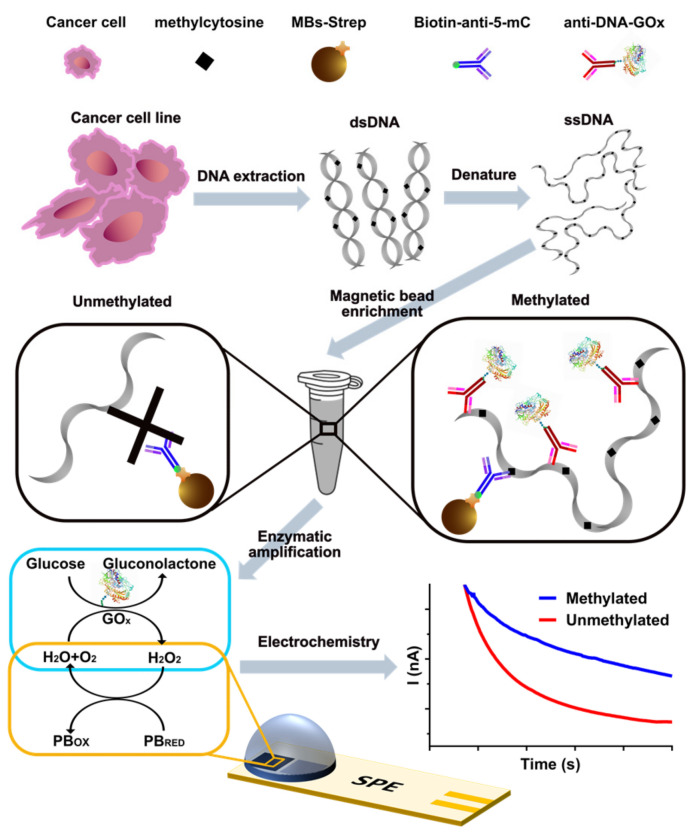
Schematic representation of the method for global DNA methylation detection. Extracted genomic DNA was denatured before being specifically captured by anti-5mC-MBs and then followed by immunorecognition of DNA using the anti-DNA-GOx antibody. GOx was employed as an enzyme label whose catalytic product was measured and the derived electrochemical results were used to determine the global methylation level.

**Figure 2 biosensors-12-00162-f002:**
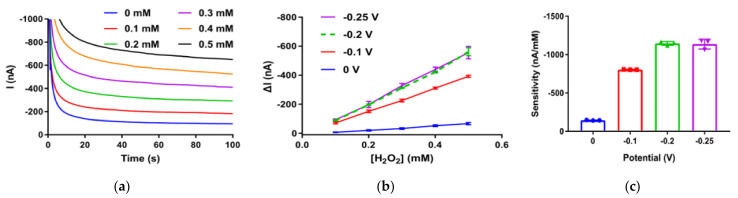
(**a**) Representative chronoamperometric response traces obtained with PB-doped electrode at –0.2 V working potential in 50 mM Tris-HCl containing different concentrations of H_2_O_2_. (**b**) Calibration plot of CA responses obtained from amperometric curves versus H_2_O_2_ concentration at different working potentials. (**c**) The corresponding sensitivity different working potentials (*n* = 3). Sensitivity of H_2_O_2_ is the slope of the calibration curve, which was plotted as H_2_O_2_ concentration versus the current value at the 100th second of the amperometric response. Error bars represent the standard deviation (SD) of three independent experiments.

**Figure 3 biosensors-12-00162-f003:**
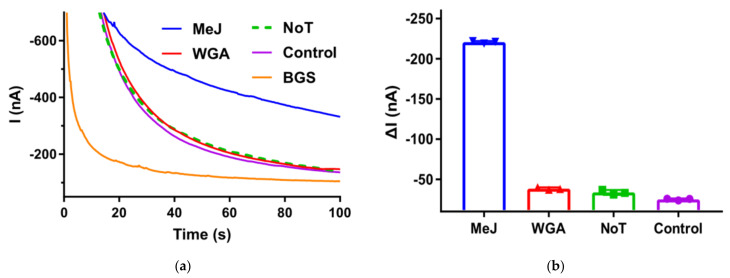
Selectivity of the proposed assay. (**a**) Representative chronoamperometric traces obtained from methylated Jurkat DNA (blue), WGA (red), no target sample (green dotted) and control (purple). (**b**) Corresponding current with different samples. Error bars estimated as the SD of three different independent experiments.

**Figure 4 biosensors-12-00162-f004:**
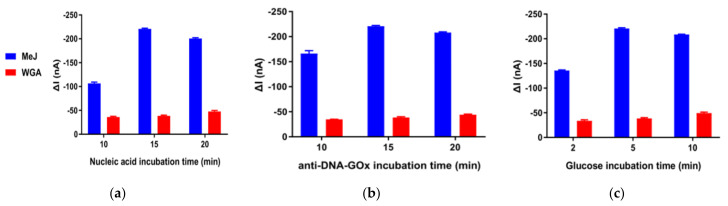
Optimization of experimental conditions. (**a**–**c**) Dependence of CA responses obtained in the presence (blue bars) and in the absence (red bars) of 10^−3^ ng/μL methylated Jurkat DNA captured by anti-5mC-MBs (**a**), later labeling them with 1/100 dilution of anti-DNA-GOx (**b**) and final enzymatic redox reaction in 10 mM glucose (**c**) with various incubation period. Error bars estimated as the SD of three different independent experiments.

**Figure 5 biosensors-12-00162-f005:**
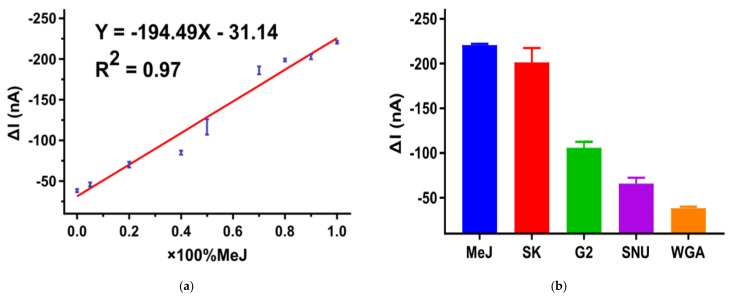
(**a**) Calibration plots of current responses obtained from amperometric curves versus 5mC at global level (*n* = 3). (**b**) CA responses comparison between different cell lines. Error bars represent the SD of three replicates.

**Table 1 biosensors-12-00162-t001:** Comparison of global DNA methylation detection method via chronoamperometry.

Biosensor	LOD	Input DNA	Assay Time	Reference
MBD magnetic bead and HRP	5% mehtylation	50 ng	2 h	[37]
Gold-DNA affinity and HRP-5mC antibody	5% methylation	25 ng	2 h	[35]
Gold-DNA affinity and 5mC functionalized nanomaterials	10% mehtylation	50 ng	1 h	[36]
Immunosensor with 5mC antibody	6.8 pM	(23–24,000) pM	45 min	[24]
Gold-DNA affinity and 5mC-GOx antibody	5% methylation	50 ng	1 h	[17]
Immuno-magnetic beads assay	4.0 pg	(14–2500) pg	45 min	[23]
Competitive immunoassays	60 pg	(0.17–7.2) ng	45 min	[25]
Immunomagnetic beads and enzymatic reaction	5% methylation	15 pg	70 min	This work

## Data Availability

Not applicable.

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
