# Peer review of "Magnetic Immunosensor Coupled to Enzymatic Signal for Determination of Genomic DNA Methylation"

_biosensors, 2022, doi:10.3390/bios12030162_

Round 1
Reviewer 1 Report
Liang et al report the work “Magnetic immunosensor coupled to enzymatic signal for de-termination of genomic DNA methylation”. This work is important and has the possibility of application in future clinical analysis towards diagnostics and prognostic predictions. The sensing strategy is quite interesting and effective, while utilizing immunomagnetic beads and enzymatic signal via electrochemical detection on SPE feasible for potential POCT applications in clinical application. Results show good sensitivity, selectivity, reproducibility and stability for genomic DNA methylation detection. From my point of view, the manuscript can be accepted after minor revision:
- Though the authors have discussed the difference between their method and other research groups’ work on global DNA methylation measurement with electrochemical detection strategy, what about those method employing bisulfite conversion and PCR amplification when comparing the detection sensitivity?
- The choice of the electron-transfer-mediator (Prussian blue) is not clearly discussed. What is the advantage compared to other candidate materials?
- How to store the sensor in application scenarios? And how to ensure the stability?
- Please revise the text carefully. There are a few typos along the manuscript.
Reviewer 2 Report
The manuscript deals with the development of an electrochemical sensor for methylated DNA detection, employing magnetic beads-based immuno reaction. The field is worthy of investigation, and the idea of coupling a sandwich-type assay for methylated DNA detection is very appealing.
Nevertheless, the concepts are described confusingly, and the English is very poor. Several terms are used in the wrong way, giving the text a particular meaning. A complete review of the English is strictly needed.
Several requests should be accomplished before the publication:
- The title is confusing, saying that the sensor is "magnetic". I suggest a rewriting
- In figure 1, the authors should check the red-ox reaction focusing on the Hydrogen Peroxide. Furthermore, in the picture, the anti-DNA GOx should be connected to the gray zone of the DNA, and the Biotin anti 5mC to the methylcytosine black spot.
- Concerning the SPE, it is not clear why the counter electrode has not been printed: the electrochemical measure is a chronoamperometry, and it usually requests the CE's presence.
- In section 3 the authors forgot to remove the template
- Section 3.1 (and in the whole manuscript): please write the error values of the data in the right way. (e.g. 143.82 ± 3.61 is not correct).
- In figure 2b for y-axis (and in the whole manuscript), please change ∂ with capital letter (namely Δ). For x axis, please write the number 2 as subscript. The same in fig. 2 caption
- Section 3.3: The optimization step is unclear, and confusing. Please rewrite it. I suggest deleting the "ratio bars" and the "wga" bars, because these experiments have already been performed for the "selectivity of the assay" section. I suggest depicting these graphs only with the blank bar (0 mM) and the 10^-3 µg/mL of MeJ
- Section 3.4: Please add a table to compare the sensor's performances with the one already in literature. Moreover, please write the calibration curve equation in the main text and add the standard deviation of the slope and intercept.
- Section 3.5: The calculation of the methylated percentage from the three HCC cell lines should be explained deeply: the sentence "by comparing data in figure 5a and figure 5b" is not analytically right
- The conclusion should be well explained and completely reorganized and rewritten. Please pay attention because some template sentences are still in the final manuscript!
Round 2
Reviewer 2 Report
I Would like to thank the authors for answering the questions. Nevertheless, some parts still need to be fixed.
Section 3.1: standard deviations of sensitivities have been deleted and not written in the right way as requested (please double-check the significant figures). Also, the standard deviations of the slope and intercept of the calibration curve are not analytically correct (please double-check the significant figures). Finally, the R2 value should be written with three digits after the 0.
